# Feature Selection Algorithms as One of the Python Data Analytical Tools †

**Nikita Pilnenskiy ‡ and Ivan Smetannikov *,‡** 

Information Technologies and Programming Faculty, ITMO University, 197101 St. Petersburg, Russia; ndpilnenskii@itmo.ru

* Correspondence: ismetannikov@itmo.ru
† This paper is an extended version of our paper published in Proceeding of the 25th conference of FRUCT association.
‡ Current address: Kronverkskiy pr.49, 197101, St.Petersburg, Russia.

**Abstract:** With the current trend of rapidly growing popularity of the Python programming language for machine learning applications, the gap between machine learning engineer needs and existing Python tools increases. Especially, it is noticeable for more classical machine learning fields, namely, feature selection, as the community attention in the last decade has mainly shifted to neural networks. This paper has two main purposes. First, we perform an overview of existing open-source Python and Python-compatible feature selection libraries, show their problems, if any, and demonstrate the gap between these libraries and the modern state of feature selection field. Then, we present new open-source scikit-learn compatible ITMO FS (Information Technologies, Mechanics and Optics University feature selection) library that is currently under development, explain how its architecture covers modern views on feature selection, and provide some code examples on how to use it with Python and its performance compared with other Python feature selection libraries.

**Keywords:** machine learning; feature selection; open-source library; Python

## 1. Introduction

The "curse of dimensionality" is one of well-known machine learning problems, as described in [1]. Basically, it states that, when the dimensionality increases, the volume of the search space increases so fast that the data become sparse. Nowadays, with the growth of data volumes and increasing effectiveness of neural networks, this problem has faded away from various fields, but it still stands in several high-dimensional data domains, namely medical care, social analysis, and bioinformatics [2–5]. For such domains, the number of objects is relatively small while the number of features can be up to several hundreds of thousands, thus resulting in object space sparsity and model overfitting.

This problem was solved with mathematical statistics [6], but nowadays it is one of the main fields of machine learning, called dimensionality reduction. The problem of selecting features from existing ones is called feature selection while feature extraction builds a projection to new feature space from the old one [7]. Unfortunately, in bioinformatics and medicine, only feature selection is applicable, as for these domains it is important to retain original features semantics for better understanding of undergoing processes [8]. Nevertheless, both approaches can reduce the feature set and increase the quality of the resulting models.

With the growth of the computational power, the neural networks approach became really powerful in many machine learning applications. As most of the libraries designed to work with neural networks were programmed in the Python language, it became de facto the international

standard for neural network research [9,10]. As the popularity of machine learning grew, more researchers were attracted to this field; since neural networks were a huge attraction point in the last years, most of the modern machine learning researchers are using Python as their main programming language. These factors resulted in a huge gap between Python machine learning libraries and libraries on other languages. Nearly all machine learning fields that are not closely tied with neural networks are not properly covered with programming libraries in Python. In this paper, we cover the main existing open-source Python feature selection libraries, show their advantages and disadvantages, and propose our own ITMO FS [11]. In addition, a comparison with Arizona State University feature selection library [12] and scikit-learn feature selection module [13] is presented. This paper has two main purposes: first, to perform an overview of existing open-source Python feature selection libraries and compare them and, second, to present the open-source ITMO FS library and some comparisons of its performance.

The rest of this paper is organized as follows, Section 2 reviews existing ways for feature selection algorithms categorization. Section 3 offers a survey of existing feature selection Python libraries and their analysis. Section 4 describes existing feature selection libraries that are available in other languages, but still can be easily adapted to Python with provided code interfaces. Section 5 contains a description of the proposed ITMO FS library and its comparison with libraries surveyed in the previous section. Section 6 has some code samples for better understanding of ITMO FS library architecture and compares its performance with some other libraries. Section 7 contains the conclusion.

## 2. Background

To better understand how the modern feature selection library should be designed and what should be included in it, we have to present available types of feature selection algorithms. In this section all main categories of existing feature selection algorithms are presented.

Generally speaking, the feature selection problem can be formalized as follows [14]: For a given dataset $D$ with $M$ objects described with feature set $F$, $|F| = N$ we need to find some optimal subset of features $F^*$, $F^* \subseteq F$ in terms of some optimization of $C$ criteria.

### 2.1. Traditional Feature Selection Algorithms Categorization

Traditionally, feature selection methods were divided into three main groups: wrapper, filter, and embedded methods [15]. Wrappers try to build optimal feature subset by the evaluation of the quality measure $Q_c$ for the predefined machine learning algorithm:

$$F^* = \underset{F' \subseteq F}{argmax}\ Q_c(F'),$$

where $C$ is a machine learning model and $Q$ is the quality measure for the model. For this, a wrapper algorithm works iteratively; on each step, it takes some feature subset and passes it to the model and then, depending on the model quality, it decides to pick another subset or stop the process. The picking procedure and the optimization stopping criteria basically define the wrapper algorithm. The main problem of this approach is that it is too slow for high-dimensional datasets as the number of possible subsets is equal to $2^N$ and on each step we need to build a model to evaluate the subset quality.

Embedded methods usually use some intrinsic properties of the classifier to get features subset. Feature selection with random forest [16] is an illustrative example of such approach, where the out of bag error for each feature on each tree is aggregated into resulting feature scores and features that most often result in the elimination of bad classification results. Some of these methods can work even with really high-dimensional data, but their main restriction is the model itself. Basically, features that were selected with one model can result in bad performance if they are used for another model.

Filters are the third traditional group of feature selection algorithms. Instead of evaluating the feature sets with some models, they take into consideration only intrinsic properties of the features themselves. If a filter does not consider any dependencies between features themselves, thus assuming

that they are independent, it is called univariate, otherwise it is multivariate. For multivariate filters, the problem is stated as follows:

$$F^* = \underset{F' \subseteq F}{argmax}\ \mu(F'),$$

where $\mu$ is the feature subset quality measure. On the other hand, for the univariate filters, the feature selection problem is stated without optimization. Instead, every feature is evaluated with feature quality measure $\mu$ (which for this case should be defined only on $f_i \in F$, but not for the whole set of features subsets) and then some cutting rule $\kappa$ is applied.

*2.2. Hybrid and Ensembling Feature Selection Algorithms*

Nowadays, most scientists distinguish separate groups of feature selection algorithms: hybrids and ensembles. Basically, these types of algorithms implement consecutive and parallel approaches to combine feature selection algorithms.

Hybrid feature selection algorithms try to combine traditional approaches consecutively. This is a powerful compromise between different traditional approaches. For example, to select features from high-dimensional dataset, a filter as a first step can be applied to drastically reduce feature set and after that a wrapper can be applied to the output feature set to get maximum quality from the extracted features.

Alternatively, ensemble feature selection algorithms combine several feature selection algorithms in parallel to improve their quality or even get better stability of selected feature subsets. Ensemble feature selection algorithms work either with separately selected features subsets or with the models, built on them [17].

*2.3. Feature Selection Algorithms Categorization by Input Data*

Some researchers categorize feature selection algorithms depending on the input data types. Basically, all data can be divided into streaming and static data. Streaming data can be divided into two big groups: data stream, when new objects are added consecutively, and feature stream, when new features are added to the dataset.

For static data, which are more conventional for traditional feature selection algorithms, some researchers [18] categorize them into: similarity based, information theory based, sparse learning based, statistical based methods, and others. Similarity based methods build an affinity matrix to get feature scores. As they are kind of univariate filters in traditional classification, they do not take into consideration any model and do not handle feature redundancy. Information theoretical algorithms work the same way as similarity based ones but also utilize the concept of "feature redundancy". Sparse learning based feature selection algorithms embed feature selection into a machine learning algorithms, working with weights of features. Statistical based feature selection models use statistical measures for features filtering, thus working exactly as most filtering methods in traditional interpretation. This categorization does not include wrappers, thus they are usually categorized as "other".

## 3. Overview of Existing Python Feature Selection Libraries

This section contains descriptions of existing open-source libraries and repositories with feature selection algorithms in Python.

*3.1. Default Scikit-Learn Feature Selection*

The scikit-learn library is de facto the most commonly used machine learning library for Python nowadays. It is so popular that nearly all other libraries and frameworks that were developed in recent years are scikit-learn compatible and easy to use with it. Nevertheless, the feature selection module of this library [13] looks really empty as it implements only several filters and wrappers. A comparison of its contents with other libraries can be found in Table 1 in the column named "SKL".

**Table 1.** Comparison of different Python FS libraries contents. Shaded box means that the algorithm is implemented in the corresponding library. ITMO stands for ITMO FS library presented in this paper, ASU for Arizona State University library, SKL for scikit-learn library, FES for FES book support code and MLR for mlr3 library.

| Algorithm | ITMO | ASU | SKL | FES | Weka | Caret | MLR | Matlab |
|---|---|---|---|---|---|---|---|---|
| Correlation | ▨ | | | | ▨ | | | |
| InformationGain | ▨ | | | | ▨ | | ▨ | |
| GiniIndex | ▨ | | | | ▨ | | | |
| F-ratio | ▨ | | | | | | | |
| fit criterion | ▨ | | | | | | | |
| MRMR | ▨ | | | | | | | ▨ |
| VDM | ▨ | | | | | | | |
| SymmUncertainity | ▨ | | | | ▨ | | | |
| AddDel | ▨ | | | | | | | |
| MeLiF | ▨ | | | | | | | |
| CIFE | | ▨ | | | | | | |
| CMIM | | ▨ | | | | | | |
| DISR | | ▨ | | | | | | |
| FCBF | | ▨ | | | | | | |
| ICAP | | ▨ | | | | | | |
| JMI | | ▨ | | | | | | |
| MIFS | | ▨ | | | | | | |
| MIM | | ▨ | | | | | | |
| FisherScore | | ▨ | | | | | | |
| ReliefF | ▨ | | | | ▨ | | | ▨ |
| TraceRatio | | ▨ | | | | | | |
| LL_L21 | | ▨ | | | | | | ▨ |
| LS_L21 | | ▨ | | | | | | ▨ |
| RFS | | ▨ | | | | | | |
| ChiSquare | ▨ | ▨ | ▨ | | | | ▨ | |
| T_score | | ▨ | | | | ▨ | | |
| AlphaInvesting | | ▨ | | | | | | |
| GraphFS | | ▨ | | | | | | |
| GroupFS | | ▨ | | | | | | |
| TreeFS | | ▨ | | | | | | |
| DecisionTreeBackward | ▨ | | | | | | | |
| DecisionTreeForward | | ▨ | | | | | | |
| SVMbackward | ▨ | | | | | | | |
| SVMforward | | ▨ | | | | | | |
| LapScore | | ▨ | | | | | | |
| SPEC | | ▨ | | | | | | |
| MCFS | | ▨ | | | | | | |
| NDFS | | ▨ | | | | | | |
| UDFS | | ▨ | | | | | | |
| ANOVA based | | | ▨ | | | ▨ | | |
| RFE | ▨ | | ▨ | | | ▨ | | |
| FPR | | | ▨ | | | | | |
| FDR | | | ▨ | | | | | |
| FWE | | | ▨ | | | | | |
| ROCfilter | | | | ▨ | | | | |
| Simulated Annealing | | | | ▨ | | ▨ | | |
| MOSS | ▨ | | | | | | | |
| MOSNS | ▨ | | | | | | | |
| Fehner | ▨ | | | | | | | |
| Backward Selection | ▨ | | | | ▨ | | ▨ | |
| Sequential Forward Selection | ▨ | | | | | | | ▨ |
| Filter ensemble | | | | | | | ▨ | |
| Stepwiseglm | | | | | | | | ▨ |
| Stepwiselm | | | | | | | | ▨ |

### 3.2. Scikit-Feature (Arizona State University)

Scikit-feature is a Python open-source feature selection library developed by the Data Mining and Machine Learning Lab at Arizona State University [12,19]. Nowadays, it is the biggest Python feature selection library that exists; it includes about 40 different feature selection algorithms. Moreover, it is completely scikit-learn compatible and easy to use. However, there are two main issues related to this library. Firstly, its development was stopped around two years ago, and at this stage the library still does not have many feature selection algorithms, especially of hybrid and ensemble types. Secondly, it is built upon feature selection algorithms categorized by the input data. This results in some issues of compatibility with theoretical basics of some algorithms and puts many limitations on the possible development of the algorithms. An example of such limitations is presented in Listing 1.

**Listing 1.** Usability comparison of ASU and ITMO FS libraries on Basehock dataset.

```
1  from skfeature.function.statistical_based import gini_index
2
3  data, target = basehock['X'], basehock['Y']
4  features = gini_index(data, target)
5  k_best = lambda x, k: [key[0] for key in x[:k]]
6  k_best(dict(zip([i for i in range(len(features))], features)), 10)
7
8
9  from ITMO_FS.filters import gini_index, select_k_best, UnivariateFilter
10
11 data, target = basehock['X'], basehock['Y']
12 filter=UnivariateFilter(gini_index, select_k_best)
13 filter.run(data, target)
```

As could be seen from this example, on Line 4, this library only extracts features list with measures, but does not consider taking cutting rule as an additional input. Of course, we can implement the required cutting rule directly in the code as presented on Line 5 and then apply it as on Line 6, but this is not a user-friendly approach, as without any template limitations during the implementation process a user may incorporate many additional errors into the code. A comparison of this library contents with other libraries can be found in Table 1 in the ASU column.

### 3.3. Boruta Methods

The Boruta methods repository [20] consists of the Python implementation of the Boruta R package that is also scikit-learn compatible. This repository is not included in Table 1 as it contains Boruta methods only and nothing else, unlike other libraries.

### 3.4. MLFeatureSelection Library

The MLFeatureSelection [21,22] library contains some heuristic feature selection algorithms based on certain machine learning methods and their evaluation techniques, such as sequence selection, importance selection, and coherence selection. This library contains these methods only, thus it is not added to the comparison in Table 1. Nevertheless, it has achieved impressive results in Rong360, JData-2018, and IJCAI-2018 competitions.

### 3.5. FES Book Support Code

This repository [23] contains sample codes for the text "Feature Engineering and Selection: A Practical Approach for Predictive Models" by Kuhn and Johnson and published by Chapman & Hall/CRC (who hold the copyright). Most of the book contents consists of feature extraction

algorithms, but the last three chapters are mainly focused on the feature selection ones, namely RFE, ROC filter, simulated annealing and some basic implementations of genetic algorithms.

*3.6. ReBATE Algorithm*

As the most popular library, the ReBATE repository [24] contains several RelieF-based method implementations that fit with scikit-learn pipeline structure. As this library also basically contains only one method, it is not included in the comparison in Table 1.

*3.7. MLxtend*

MLxtend [25] is a Python library of useful tools for the day-to-day data science tasks. The feature selection part is represented by several sequential wrappers with different optimization types and exhaustive models.

## 4. Python Compatible Feature Selection Libraries

*4.1. Weka in Java*

Since Weka [26] is one of the biggest machine learning libraries, it includes many valuable feature selection methods. Feature selection has various filter methods and wrappers.

*4.2. Caret in R*

Caret (Classification And REgression Training) [27] is a set of functions that attempts to streamline the process for creating predictive models for R language. It has wrapper methods based on recursive feature elimination, genetic algorithms and simulated annealing. For filter methodology, caret implements univariate filer, which can take any scorer function.

*4.3. MLR in R*

MLR [28] is one of the most comprehensive machine learning libraries for R language. It contains not only feature selection, but also different models for forecasting, clustering, image processing, and some other problems. A feature selection module has a filter, a wrapper, and an ensemble of filters in feature selection section.

*4.4. Feature Selection in Matlab*

Math language Matlab [29] has its own feature selection module in its core code. It has some filters, a sequential wrapper, and about a dozen embedded algorithms.

## 5. ITMO FS library Architecture and Comparison

The main issues observed in the libraries presented in Sections 3 and 4 libraries are as follows:

- does not support customizable cutting rules for filters (nearly all of them);
- does not support ensemble or hybrid feature selection algorithms (which are actually the most commonly used in recent years, nearly all of them);
- has high requirements and long delays for new algorithms to be added (scikit-learn);
- uses input data paradigm as basis for architecture, as described in Section 2.3, which is not usually considered (ASU); and
- programmed in other languages so has to be either adopted or wrapped with additional computational costs.

Thus, the ITMO FS library was decided to be developed.

In Figure 1, the ITMO FS library architecture is shown. It contains filters, wrappers, hybrid, and embedded parts, each of which represents a set of methods from traditional classification.

Unfortunately, ensembles are not currently presented in the library scheme as they are still under development. This categorization was inspired by traditional feature selection categorization. All filters are generally described as shown in Section 2.1, thus they need to have some feature quality measure and a cutting rule. The wrapper module contains several wrapper algorithms. Each of them takes an estimator as input parameter and improves the feature set regarding model performance. The embedded part contains MOSS, MOSNS, and RFE algorithms, each of which selects the feature set according to its importance to the model. The hybrid module for now contains only MeLiF algorithm that was developed by ITMO University [14].

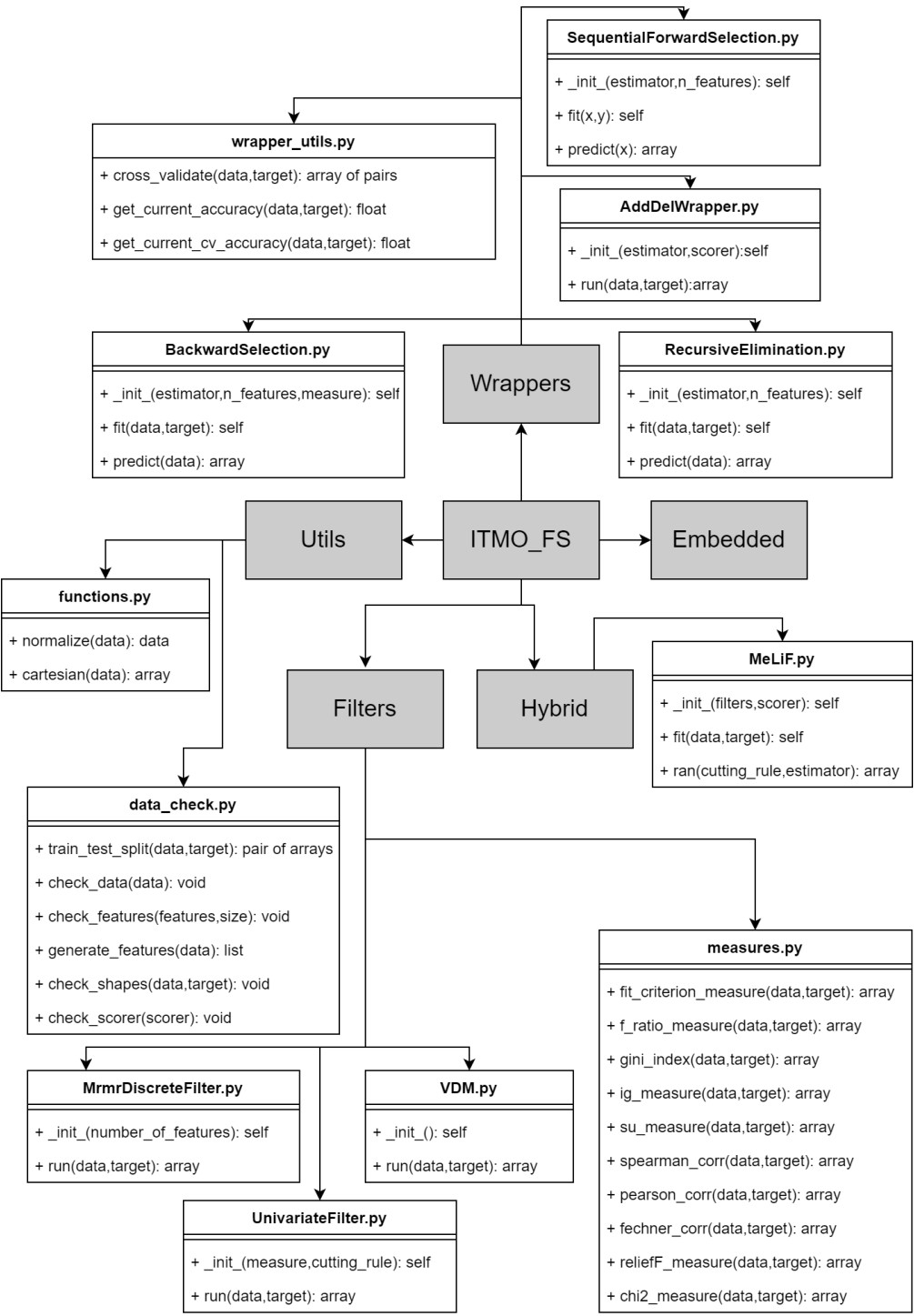

**Figure 1.** ITMO FS library architecture.

The only prerequisite for library is numpy package; the library is developed with Python 3 language. The current version of the library contents is shown below and also it could be found on the architecture scheme in Figure 1:

- Filters

    - UnivariateFilter—class for constructing custom filter (UnivariateFilter.py);
    - MRMR—class for specific filter (MRMRFilter.py);
    - VDM—class for specific filter (VDM.py);

- Hybrid

    - MeLiF—class for basic MeLiF (MeLiF.py);

- Wrappers

    - AddDel—class for ADD-DEL wrapper (AddDelWrapper.py);
    - Backward elimination—class for backward elimination wrapper (BackwardSelection.py);
    - Sequential forward selection—class for sequential forward selection (SequentialForwardSelection.py);

- Embedded

    - MOSNS—Minimizing Overlapping Selection under No-Sampling (MOSNS.py);
    - MOSS—Minimizing Overlapping Selection under SMOTE (MOSS.py);
    - Reccursive Feature Elemenation—class for RFE method (RecursiveElimination.py).

Spearman correlation, Pearson correlation, information gain, Gini index, F-ratio, fit criterion, and symmetric uncertainty are custom measures that are stored with cutting rules in the Filter file. In the future, when we add more basic measures to the library and some more exotic cutting rules, they could be easily added to the library through this file. MeLiF is one of the hybrid methods and it aggregates the result of filter measures and wraps an estimator, greedy optimizing their ensemble weights. The pipeline is displayed in Figure 2. This algorithm builds a linear combination of the input filters quality measures with some $\alpha$ coefficients and then tunes them in the optimization cycle of the model $C$ with some quality measure $Q_C$ (for example, $F_1$ score). In this case, a single cutting rule $\kappa$ is applied for the whole combination.

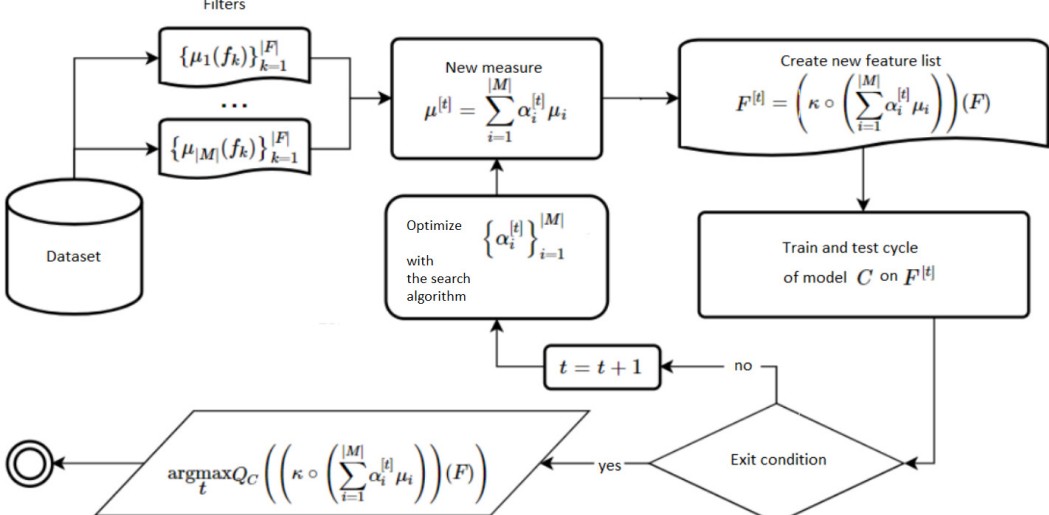

**Figure 2.** Hybrid algorithm MeLiF pipeline.

As shown in Table 1, the Arizona State University feature selection library has the most implemented algorithms in comparison with others, but this library was updated last time about two years ago. Moreover, it does not have some basic feature selection algorithms such as information gain filter, AddDel, RFE, ROC filter, forward and backward selection, and some others. Scikit-learn feature selection module seems to have some rarely used methods and the general volume of presented algorithms is rather small: it has only six of them. Feature extraction and selection book contains some examples of exotic methods; nevertheless, there are many different feature extraction methods there. Thus, although it is quite useful for dimensionality reduction in general, it is not usable in practice for feature selection. On the right of the double line in Table 1, Python-compatible feature selection libraries from other languages are shown. None of them has good enough coverage of existing feature selection algorithms. Moreover, as they were not developed in Python from the beginning, their usage requires some additional actions and their computational time is somewhat slower. Development of ITMO Feature Selection library started in the spring of 2019, and at this stage the library has already implemented almost all basic architectural elements; it is scikit-learn compatible and user-friendly. It contains some basic algorithms used in practice as well as some less common methods.

## 6. ITMO FS Library Usage Examples and Performance Tests

For all code samples, imports from the Listing 2 are required.

**Listing 2.** Imports for this section experimental setup.

```
1  import time
2  from skfeature.function.statistical_based import gini_index
3  from sklearn.datasets import load_iris
4  from sklearn.feature_selection import SelectKBest
5  from sklearn.linear_model import LogisticRegression
6  from sklearn.metrics import f1_score
7  from sklearn.svm import SVC
8  from ITMO_FS.filters import *
9  from ITMO_FS.hybrid.Melif import Melif
10 from ITMO_FS.wrappers.AddDelWrapper import *
11 from ITMO_FS.wrappers.BackwardSelection import *
```

During testing, we used the following software versions: sklearn 0.21.3, MLFeatureSelection 0.0.9.5.1, skrebate 0.6, python 3.6.9, numpy 1.17.3, scipy 1.3.1, Boruta 0.3, and ReliefF 0.1.2. In the test_filter function, an example of the filter class with Spearman correlation and "Best by value" cutting rule that equals 0.99 is shown. This notation means that all features which have score higher than 0.99 will be selected. A possibility to customize the cutting rule such as in this example is only supported by the ITMO FS library and is shown in Listing 3 For this example, iris [30] dataset was chosen.

**Listing 3.** Filter usage on the iris dataset with the custom cutting rule.

```
1  def test_filter(self):
2      filtering = UnivariateFilter(spearman_corr, select_best_by_value(0.99))
3      data, target = load_iris(True)
4      res = filtering.run(data, target)
5      print(data.shape, '--->', res.shape)
```

In Listing 4, an example of UnivariateFilter class with Pearson correlation and "Best by value" cutting rule that equals 0.0 is shown. A possibility to use lambda functions for cutting rules is only supported by the ITMO FS library. This example uses orlraws10P dataset [31].

**Listing 4.** Filter usage on the ORL10 dataset with the custom lambda cutting rule.

```
1  def test_filter(self):
2      value=0
3      rule=lambda scores:[k for k,v in scores.items() if v>value]
4      filtering = UnivariateFilter(pearson_corr, rule)
5      data=scipy.io.loadmat(orlaws)
6      data, target = data['X'], data['Y']
7      res = filtering.run(data, target)
8      print(data.shape, '——>', res.shape)
```

In Listing 5, the UnivariateFilter class with Pearson correlation and "K best" cutting rule with six best features selected is shown on Line 5. It is compared with scikit-learn SelectKBest usage on Line 9. The test dataset here is Basehock. As can be seen from this example, ITMO FS template is more user friendly and gives an opportunity to use some other cutting rules than "K best". The testing setup for the Gini index is the same but with the usage of gini_index function instead of pearson_corr.

**Listing 5.** Comparison of the K-best cutting rule and Pearson correlation usage at ITMO FS and scikit libraries.

```
1  def test_pearson_k_best(self):
2      data, target = self.basehock['X'], self.basehock['Y']
3      start_time = time.time()
4
5      res = Filter(pearson_corr,select_k_best(6)).run(data, target)
6      print("ITMO_FS_time——%s_seconds——" % (time.time() − start_time))
7      start_time = time.time()
8
9      res = SelectKBest(pearson_corr,k=6).fit_transform(data, target)
10     print("SKLEARN_time——%s_seconds——" % (time.time() − start_time))
11     print(data.shape, '——>', res.shape)
```

In Listing 6, the AddDel (Line 4) wrapper method with basic logistic regression (Line 3) as estimator is shown. For quality measure function, $F_1$ score was chosen. Test dataset here is Basehock.

**Listing 6.** Example of Add-del wrapper usage.

```
1  def test_add_del():
2      data, target = basehock['X'], basehock['Y']
3      lr = LogisticRegression()
4      wrapper = Add_del(lr, f1_score)
5      wrapper.fit(data, target)
6      print(wrapper.best_score)
```

In Listing 7, the backward selection wrapper (Line 4) method with logistic regression (Line 3) as estimator is shown. Test dataset here is Basehock. Although the BackwardSelection function is

also a wrapper, it has an additional parameter "100" that defines maximum number of features to be removed. In this case, it is a "stopping criteria" in the standard wrapper notation.

**Listing 7.** Example of Backward selection wrapper usage.

```
1  def test_backward_selection ():
2      data, target = basehock['X'], basehock['Y']
3      lr = LogisticRegression ()
4      wrapper = BackwardSelection(lr, 100, gini_index)
5      wrapper.fit(data, target)
6      print(wrapper.best_score)
```

Listing 8 shows an example of the MeLiF class with Support vector classifier (Line 7) as estimator. Chosen filters are Gini index (Line 4), fratio (Line 5), and information gain (Line 6). $F_1$ score was chosen as quality measure for MeLiF (Line 7). All input parameters of the ensemble including the cutting rules can be tuned.

**Listing 8.** Example of ensemble algorithm MeLiF usage.

```
1   def test_melif ():
2       data, target = basehock['X'], .basehock['Y']
3       _filters = [
4       UnivariateFilter (gini_index, select_best_by_value (0.4)),
5       UnivariateFilter (F_ratio (data.shape[1]), select_best_by_value (0.6)),
6       UnivariateFilter (ig_measure, select_best_by_value (−0.4))]
7       melif = Melif(_filters, f1_score)
8       melif.fit(data, target)
9       estimator = SVC()
10      melif.run(select_k_best(50), estimator)
```

Table 2 shows time comparison between ITMO Feature Selection library and scikit-learn feature selection module at selecting k best features by Gini index and Pearson correlation. Basehock dataset [32] has 1993 samples and 4862 features, COIL20 dataset [33] has 1440 samples and 1024 features, and orlraws10P has 100 samples and 10,304 features. The comparison was performed on Intel i7-6700HQ, 4 cores, 2.6 GHz,16 GB DDR4-2400, HDD Toshiba MQ01ABD100, 5400 rpm.

**Table 2.** Comparison of sklearn feature selection module and ITMO FS library computational time in seconds.

| Dataset | Library | Pearson | Gini Index |
|---------|---------|---------|------------|
| Basehock | ITMO FS | 0.129 | 0.373 |
| | SKLEARN | 0.122 | 0.379 |
| COIL20 | ITMO FS | 0.020 | 0.044 |
| | SKLEARN | 0.025 | 0.057 |
| orlraws10P | ITMO FS | 0.019 | 0.049 |
| | SKLEARN | 0.017 | 0.049 |

As shown in Table 2, the ITMO FS library has approximately the same computational time for Pearson correlation coefficient and Gini index filters as in the scikit-learn feature selection module. As these filters are not implemented in scikit-learn, we have put our customized measures in it for this reason.

Listing 9 shows code usage comparison between ASU (Arizona State University) Feature Selection library and ITMO University library. This example was used to estimate execution time of Gini index and F-score index for both libraries. The result of this comparison can be found in Table 3.

**Listing 9.** Time comparison between ASU feature selection and ITMO FS.

```
1  def test_arizona():
2      data, target = coil['X'], coil['Y']
3      start_time = time.time()
4      features = gini_index.gini_index(data, target)
5      print("ARIZONA_time ___ _%s_seconds ___ " % (time.time() − start_time))
6      start_time = time.time()
7      features = gini_index(data, target)
8      print("ITMO_time ___ _%s_seconds ___ " % (time.time() − start_time))
9      start_time = time.time()
10     features = f_score.f_score(data, target)
11     print("ARIZONA_time ___ _%s_seconds ___ "% (time.time() − start_time))
12     start_time = time.time()
13     features = f_ratio(data.shape[−1])(data, target)
14     print("ITMO_time ___ _%s_seconds ___ "  % (time.time() − start_time))
```

As shown in Table 3, the ITMO FS library has a slightly bigger computational time for F-score index and far better time for Gini index than the Arizona State University feature selection library. As ASU library does not provide support for cutting rules in the traditional way, we ran intrinsic measures of ITMO FS algorithms separately from cutting rules for proper comparison.

**Table 3.** Comparison of ASU feature selection library and ITMO FS library computational time in seconds.

| Dataset | Library | F-score index | Gini index |
|---------|---------|---------------|------------|
| Basehock | ITMO | 1.084 | 0.376 |
|          | ASU  | 0.116 | 1.507 |
| COIL     | ITMO | 1.081 | 0.048 |
|          | ASU  | 0.048 | 253.257 |
| ORL      | ITMO | 5.793 | 0.058 |
|          | ASU  | 0.028 | 97.622 |

## 7. Conclusions

In the last few years, Python programming language has become extremely popular in most machine learning applications. Unfortunately, most of the existing Python machine learning tools are neural networks oriented, which results in an increasing gap between existing and implemented methods for most classical machine learning fields.

- This paper contains an overview of existing Python feature selection libraries and libraries on other languages that are easily compatible with Python.
- This paper pProvides a description of newly created feature selection library ITMO FS. We provided a complete description of the library including program architecture design.
- This paper shows that ITMO FS library provides better view to the traditional feature selection algorithms, supports hybrids and ensembles, and in most cases is more user-friendly than other libraries. Moreover, some of the provided test showed that ITMO FS library works more quickly than the biggest Python feature selection ASU library on some algorithms.
- This paper provides code samples for better understanding of library usage.

In the future, we plan to implement all algorithms mentioned in Table 1, more modern algorithms of feature selection as well as more classical ones, and then move to implementation of different ensembling and hybrid algorithms. In this perspective, we will also add meta-learning approaches for easier selection of feature selection algorithms and their tuning.

**Author Contributions:** Conceptualization, I.S.; Data curation, N.P.; Formal analysis, I.S.; Funding acquisition, I.S.; Investigation, N.P.; Methodology, I.S.; Resources, I.S.; Software, N.P.; Supervision, I.S.; Visualization, N.P.; Writing—original draft, N.P.; and Writing—review and editing, I.S. All authors have read and agreed to the published version of the manuscript.

**Funding:** This research was funded by the Government of the Russian Federation, Grant 08-08 and Research and Development grant No. 619421 "Development of the Python open-source feature selection library".

**Conflicts of Interest:** The authors declare no conflict of interest.

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
