# Peer review of "Feature Selection Algorithms as One of the Python Data Analytical Tools"

_futureinternet, doi:10.3390/fi12030054_

Round 1

Reviewer 1 Report

Dear Authors

Best regards

Reviewer 2 Report

The developed project is interesting, but the paper has several drawbacks. First of all, it is not clear why a complete new library should be developed. As the authors suggest, the scikit library is widely adopted and has several (not all) feature extraction algorithms. So it should be obvious to integrate new features inside an existing popular project instead of reinventing the wheel from scratch. Second problem: the paper is a mere manual (looks more like a README) rather than a scientific publication. There is no explanation of the internal mechanisms adopted to develop the tool, and it is not clear what advantages it gives w.r.t. the scikit learn library (apart from implementing more feature extraction algorithms, but refer to my previous comment).

Other comments are reported in the following (L = line).

L14: Please, introduce the concept of "curse of dimensionality". Since you base your initial discussion with that, it would be nice to have a summary on that right away.

L129: There are two problems here. First, I do not understand the example. More in particular, your objective here is to demonstrate the limitations of the Scikit-feature library. However, the script uses the ITMO_FS library. Also the observations you made about the outcomes of the script are not clear. Second problem: use an algorithmic environment (if you use Latex, as it looks like) to better illustrate the algorithms (easier and immediate references to code lines).

L152: Not sure if a book review fits in a Section called "Survey of existing Python feature selection libraries". Have the described algorithms been implemented and publicly available?

L184: "On image 1" -> In Figure 1

L196: "on picture 1" -> In Figure 1

L217: "Pipeline displayed below on the Fig. 2:". Explain the pipeline.

L219: "[...] but last time this library was updated about 2 years ago". Clearly, this could generate problems with integrations into newer Python versions. However, such a critique does not relate to the goodness, applicability, or usability of that library (a "simple" porting to a newer Python version and some bug fixes could make it usable again). Motivate better by expanding "it doesn’t have some basic feature selection algorithms".

L220: "Scikit-learn feature selection module seems to have some rarely used methods and the general volume of presented algorithms is rather small". Quantify such a volume.

L225: "To the right from the double line you can find Python-compatible feature selection libraries". Specify you are referring to Table 1.

Table 1: Explain what shaded boxes stand for.

Section 6: This section is unreadable. Please try to apply some aesthetic adjustments.

Table 3: I think that comparing executing times w.r.t. the ASU library is not very meaningful since, as you reported, it is an abandoned library that runs on Python 2.7).

Reviewer 3 Report

- Summary evaluation:

in this paper, the authors propose the feature selection library “ITMO FS”, describe the proposed open-source library that is developed in Phyton lenguage and they provide a comparison with other two open-source feature selection libraries (Arizona State University FS and scikit-learn FS).

The review of python FS libraries, the description of the proposed ITMO FS library, and the comparison of these libraries are done well.

- specific comments:

(Major comments)

With regard to the times shown in tables 2 and 3, it is necessary to describe the hardware / software environment with which they were obtained.

(Minor revision)

The introdution could highlight the usefulness of FS algorithms in the history of research, and the need to have high-performance FS algorithms for new research trends such as CNN used to extract large set of features (for exemple mention: DOI: 10.3390/app9020307 and 10.3390/app9030408)

Round 2

Reviewer 1 Report

I suggest an acceptance. 

Author Response

Thank you, please close the review.

Reviewer 2 Report

The paper is a mix between a review and a research paper. In my opinion, it isn't neither.

It is not a good survey: Considered techniques span from obsolete libraries (ASU) to code snippetts published in a book, through libraries ported from other languages that solve specific tasks (e.g., he Boruta R package). In my opinion the main objective of a survey is to select and then clarify the state-of-the-art, providing sound and meaningful comparisons. The final message of the "survey section" is inexistent.

It is not a good research paper: Motivations are completely missing. The authors fail to explain what is the main reason that pushed them to start developing a new library, and they also fail to explain why a new library is actually needed (as per the "survey part", there are a lot of existing solutions, that put together cover most available techniques). Having all existing algorithms described in Python in a single library cannot be considered a scientific publication (this is also in response to the second answer in the rebuttal). It totally lacks any novelty and does not contribute to the state-of-the-art by any means. Finally, I'd like to stress again that comparing libraries built with different technologies does not make sense (e.g., consider table 2 and table 3: no difference between ITMO and SKLEARN. On the opposite, there is a significant difference between ITMO and ASU, the latter being an obsolete Python 2 library. The difference is in the adopted technology, so I don't understand how it can be assumed as a contribution).

Other main comments:

  • I already suggested to improve the representation of code in the previous review, but I'd like to be more explicit: include line numbers in any listing.
  • Section 6 is unintelligible. I don't get a sense out of it.

Other minor comments (L = line).

L135-136: "As it could be seen from this example, at line 2 this library only extracts feature list with feature measures, but does not consider taking cutting rule as an additional input". I'm not sure what the authors are referring to. Line 2 reports "data , t a r g e t = basehock [ ’X ’ ] , basehock [ ’Y ’ ]" which is a simple variable assignment. Moreover, the listing is not complete since "basehock" is an undefined entity.

L135-140: I don't get the point. The explanation is also not very clear.

L212: "In the future when we will be adding more basic measures to the library and some more exotic cutting rules they could be easily added to the library through this file". Not sure why this information is important.

Still several typos, e.g.:

L64: "For this wrapper algorithms works iterative"

L102: "when new object are added"

L117: "algorithms on Python"
